# Morphology and Phylogeny of *Scrippsiella precaria* Montresor & Zingone (Thoracosphaerales, Dinophyceae) from Korean Coastal Waters

**Hyun Jung Kim** [1,2], **Zhun Li** [3], **Nam Seon Kang** [4], **Haifeng Gu** [5], **Daekyung Kim** [6], **Min Ho Seo** [7], **Sang Deuk Lee** [8], **Suk Min Yun** [8], **Seok-Jin Oh** [2] and **Hyeon Ho Shin** [1,*]

1   Library of Marine Samples, Korea Institute of Ocean Science & Technology, Geoje 53201, Korea; guswjd9160@kiost.ac.kr
2   Laboratory of Coastal Ecology and Environment, Department of Oceanography, Pukyong National University, Yongso-ro, Busan 48513, Korea; sjoh1972@pknu.ac.kr
3   Biological Resource Center/Korean Collection for Type Cultures (KCTC), Korea Research Institute of Bioscience and Biotechnology, Jeongeup 56212, Korea; lizhun@kribb.re.kr
4   Department of Taxonomy and Systematics, National Marine Biodiversity Institute of Korea, Seocheon 33662, Korea; kang3610@mabik.re.kr
5   Department of Marine Biology and Ecology, Third Institute of Oceanography, Ministry of Natural Resources, Xiamen 361005, China; guhaifeng@tio.org.cn
6   Daegu Center, Korea Basic Science Institute (KBSI), Daegu 41566, Korea; dkim@kbsi.re.kr
7   Marine Ecology Research Center, Yeosu 59697, Korea; copepod79@gmail.com
8   Bioresources Collection and Research Team, Nakdonggang National Institute of Biological Resources (NNIBR), Sangju 37242, Korea; diatom83@nnibr.re.kr (S.D.L.); horriwar@nnibr.re.kr (S.M.Y.)
*   Correspondence: shh961121@kiost.ac.kr; Tel.: +82-55-639-8440; Fax: +82-55-639-8429

**Abstract:** The dinoflagellate genus *Scrippsiella* is a common member of phytoplankton and their cysts are also frequently reported in coastal sediments worldwide. However, the diversity *of Scrippsiella* in Korean waters has not been fully investigated. Here, several isolates of *Scrippsiella precaria* collected from Korean waters and germinated from resting cysts were examined using light and scanning electron microscopy. The resting cysts were characterized by pointed calcareous spines and one or two red accumulation bodies, and the archeopyle was mesoepicystal, representing the loss of 2–4′ and 1–3a paraplates. Rounded resting cysts were found in culture, and an increase in spine length was observed until 8 days of development. Korean isolates of *S. precaria* had the plate formula of Po, X, 4′, 3a, 7″, 6C, 4S, 5‴, 2⁗. There were differences in the cell size and location of the red body between Korean isolates and previously described cells of *S. precaria*. In addition, the Korean isolates of *S. precaria* had two types of the 5″ plate that either contacted the 2a plate or not. Molecular phylogeny based on internal transcribed spacer (ITS) and large subunit (LSU) rDNA sequences revealed that the Korean isolates were nested within the subclade of PRE (*S. precaria* and related species) in the clade of *Scrippsiella* sensu lato, and that the PRE subclade had two ribotypes: ribotype 1 consisting of the isolates from Korea, China, and Australia, and ribotype 2 consisting of the isolates from Italy and Greece. Lineages between isolates of ribotype 1 were likely to be related to the dispersal by ocean currents and ballast waters from international shipping, and the two types of spine shapes and locations of the 5″ plates may be a distinct feature for ribotype 1.

**Keywords:** *Scrippsiella*; resting cyst; intercalary plate; precingular plate; ribotype

## 1. Introduction

The dinoflagellate genus *Scrippsiella* Balech ex A.R.Loeblich belongs to the family Thoracosphaeraceae and was established by Balech, with *Scrippsiella sweeneyae* Balech as the type species [1]. *Scrippsiella* species share the consistent plate pattern of Po, X, 4′, 3a, 7″, 6C, 5S, 5‴, 2⁗ [1–5], and based on morpho-molecular features, including the plate pattern and phylogenetic position [6], approximately 30 species are currently assigned

to the genus *Scrippsiella*, which includes some previously recorded *Calcigonellum* Deflandre species [6–11]. Most of the *Scrippsiella* species produce resting cysts with distinctive calcareous ornaments [12], but some *Scrippsiella* species, such as *Scrippsiella donghaiensis* H.Gu and *Scrippsiella enormis* H.Gu, produce noncalcareous resting cysts [13,14]. Recently, the morphological features of calcareous ornaments have been used to classify *Scrippsiella* species [13,15,16].

In previous studies, *Scrippsiella* sensu lato (s.l.) was identified based on internal transcribed spacer (ITS) sequences and large subunit ribosomal DNA (LSU rDNA) [13,15,17,18]. As *Scrippsiella* s.l. includes not only common *Scrippsiella* species but also several cyst-based genera, such as *Calciodinellum*, *Pernambugia* Janofske & Karwath, and *Naiadinium polonicum* (Woloszynskia) S. Carty, which is a freshwater species [15,18], *Scrippsiella* s.l. is not fully resolved, either phylogenetically or taxonomically. Consequently, the morphological and genetic diversity of *Scrippsiella* species from many coastal areas requires additional exploration. However, most studies on the classification of *Scrippsiella* species using morphological and genetic approaches have focused on the Mediterranean and Atlantic diversities [18,19]. Gu et al. [13,20] and Luo et al. [15] recently reported the diversity of *Scrippsiella* species from Chinese coastal areas, and the morphology and phylogeny of several *Scrippsiella* species from Korean coastal areas, such as *Scrippsiella lachrymosa* and *Scrippsiella masanensis*, have been described [21,22].

*S. precaria* Montresor & Zingone was originally described from established cultures in coastal waters of the Gulf of Naples [23], where cyst formation of this species was also reported [24]. Gu et al. [19] later provided the morphological details and elucidated the phylogenetic relationships of *S. precaria* collected from Chinese coastal waters. Cysts of *S. precaria* have also been reported in Japanese coastal waters [25,26]; however, vegetative cells and resting cysts of *S. precaria* have not yet been reported from Korean coastal areas. According to Montresor and Zingone [23] and Gu et al. [20], *S. precaria* share three anterior intercalary plates with *Scrippsiella ramonii* and *Scrippsiella irregularis*, and these species form a clade phylogenetically. However, the clade also includes *N. polonicum*, which has two anterior intercalary plates. This indicates that the arrangements of adjacent plates, possibly including precingular plates, can vary depending on the number and arrangement of anterior intercalary plates in the group, and thus, this character may be useful for the classification of species that are members of this clade.

During a study of marine dinoflagellates and resting cysts from Korean coastal areas, a *Scrippsiella*-like species and calcareous resting cysts were found. The cultures were successfully established based on the incubation of the collected vegetative cell and germination experiments. These cultures were examined using light and scanning electron microscopy, and ITS and LSU rDNA were sequenced. The results indicated that the cultures belonged to *S. precaria*. In the present study, we established the morphological details and phylogenetic positions of the Korean cultures of *S. precaria*, and then we compared them with those previously reported in other studies.

## 2. Materials and Methods

### 2.1. Sample Collection and Culture

Sediment samples were collected from a station at Jinhae-Masan Bay (34°59′36″ N, 128°40′33″ E) in July 2013 using a gravity corer. The top 2 cm of the core samples was sliced and then stored in dark and cool conditions at 4 °C prior to further analysis. Sample analysis was conducted using the panning method of Matsuoka and Fukuyo [27]; approximately 2 g of each sample was placed in a beaker, rinsed with filtered seawater, and sonicated for approximately 30 s. The sediment suspension was sieved using stainless steel screens with 125 and 10 μm mesh opening sizes. The residue on the 10 μm mesh was washed into a watch glass. By panning on the watch glass, cysts and light particles were separated from heavier sand grains and were sieved again onto the 10 μm sieve and concentrated into a final sample volume of 10 mL. To observe the cysts, 1 mL of the sample was placed in a Sedgewick Rafter chamber (Pyser-SGI, Edenbridge, Kent, UK), and the cysts of *Scrippsiella*



species characterized by numerous calcareous spines were photographed using a digital camera and then isolated under an inverted microscope.

The isolated cysts were inoculated into individual wells of 96-well tissue culture plates filled with f/2-Si culture medium (Marine Water Enrichment Solution, Sigma-Aldrich, Saint Louis, MO, USA) via micropipetting using a capillary pipette and cultured at 20 °C and ≈100 µmol photons m$^{-2}$ s$^{-1}$ cool-white illumination under a 14L:10D photocycle. Germinated cells were transferred into the wells of six-well tissue plates. After sufficient growth, the cells in the six-well tissue plates were transferred to a 30 mL culture flask containing 25 mL sterile f/2-Si medium. A monoclonal culture of the germinated cells was successfully established and deposited as strain LMBE-C28 in the Library of Marine Samples, Korea Institute of Ocean Science and Technology, Republic of Korea; however, the strain is currently not available.

On 15 March 2019, plankton samples were collected at a station in the Jinhae-Masan Bay, Korea (35°00′38″ N, 128°34′03″ E) using a 20 µm mesh plankton net. In the laboratory, a single cell of *Scrippsiella* species was isolated from the samples using a capillary pipette. The isolated cell was inoculated into a well of a 48-well tissue culture plate filled with f/2-Si culture medium and cultured at a temperature of 20 °C and ≈100 µmol photons m$^{-2}$ s$^{-1}$ cool-white illumination under a 12L:12D photocycle. The cultured cell was transferred into the wells of six-well tissue plates. After sufficient growth, the cells in the six-well tissue plates were transferred to a 30 mL culture flask containing 25 mL sterile f/2-Si medium. A monoclonal culture of *Scrippsiella* species was successfully established and deposited as strain LIMS-PS-2799 in the Library of Marine Samples, Korea Institute of Ocean Science and Technology, Republic of Korea.

### 2.2. Light Microscopy

Vegetative cells were examined and photographed at ×400 magnification using an ultra-high-resolution digital camera (DS-Ri2, Nikon, Tokyo, Japan) on an upright microscope (ECLIPSE Ni, Nikon, Tokyo, Japan). For fluorescence microscopy, approximately 1 mL of culture strain LIMS-PS-2799 was transferred to a 1.5 mL microcentrifuge tube and SYTOX® Green Nucleic Acid Stain (Molecular Probes, Eugene, OR, USA) was added at a final concentration of 1.0 µM. The cells were incubated in the dark at room temperature for 30 min and then observed using a Zeiss Filterset (emission: BP 450–490; beam splitter: FT 510) and photographed using an AxioCam MRc digital camera on an upright microscope (Axio Imager 2, Zeiss, Jena, Germany).

To observe changes in the morphology of the resting cyst, a round resting cyst produced from the culture strain LIMS-C28 was isolated and inoculated into Petri dishes (50 × 15 mm; SPL, Pocheon, South Korea) containing 10 mL of f/2-Si culture medium with a salinity of 32. The cyst was incubated on a live-cell observation system (Zeiss) at 20 °C under 100 µmol photons m$^{-2}$ s$^{-1}$ cool-white illumination in a 24L:0D photocycle. Time-lapse sequences were captured at ×400 magnification with an AxioCam MRm digital camera on an Axio Imager 2 upright microscope (Zeiss) using bright field illumination.

### 2.3. Scanning Electron Microscopy (SEM)

For the SEM, the cells were fixed with Lugol's solution for 4 h at room temperature. The fixed cells were deposited on polycarbonate membrane filters (2.0 µm pore size; Millipore, Billerica, MA, USA). The filters were rinsed twice with deionized water and dehydrated in a graded ethanol series (10–99.9% in eight steps) for 10 min per step. Filters were then critical point dried (CPD) using a critical point drying apparatus (Spi-Dry$^{TM}$ Regular Critical Point Dryer, SPI Supplies, West Chester, PA, USA) and liquid $CO_2$. After they were CPD, the filters were mounted on stubs. Finally, the samples were coated with platinum-palladium and examined using a field emission scanning electron microscope equipped with X-ray energy dispersive spectroscopy (EDS) (JSM 7600F, JEOL, Tokyo, Japan).

### 2.4. DNA Extraction and Sequencing

Genomic DNA was extracted from 1 mL of exponentially growing cultures using the DNeasy® Plant Mini Kit (QIAGEN Inc., Valencia, CA, USA). The ITS1-5.8S-ITS2 sequences for two strains LIMS-PS-2799 and LMBE-C28 were amplified using the primer pairs ITSFor and ITSRev [28]. Part of the LSU region sequence for strain LIMS-PS-2799 was amplified using the primer pairs 25F1 and R2 [29]. Polymerase chain reaction (PCR) was conducted using a thermoblock (T100™ Thermal Cycler; Bio-Rad, Hercules, CA, USA) using the following protocol: 95 °C for 4 min; 30 cycles of denaturation at 95 °C for 10 s, annealing at 52 °C for 40 s, extension at 72 °C for 1 min, and a final elongation at 72 °C for 5 min. The PCR-amplified products were confirmed using 1% agarose gel electrophoresis. The PCR products were purified with the QIAquick PCR Purification kit (Qiagen, Valencia, CA, USA). A cycle-sequencing reaction was performed using the ABI PRISM® Big Dye™ Terminator Cycle Sequencing Ready Reaction Kit (Applied Biosystems, Foster City, CA, USA).

### 2.5. Alignment and Phylogenetic Analyses

Sequences were viewed and assembled in DNABaser version 4.36 (http://www.dnabaser.com). Contigs were aligned using Mafft v6.624b online version 7 (http://mafft.cbrc.jp/alignment/server/) [30] and the Q-INS-I option was used to consider rRNA secondary structures. The GTR+I+G substitution model was selected using the Akaike information criterion, as implemented in jModelTest version 2.1.4 [31]. For the analysis of ITS1-5.8S-ITS2 sequences, the dataset contained 51 taxa and consisted of 621 characters (including gaps inserted for alignment). The armored dinoflagellate *Ensiculifera* aff. *loeblichii* (HQ845328) and *Pentaphasodinium dalei* (JX262496) were used as outgroup taxa. For the analysis of LSU sequences, the dataset contained 44 taxa and consisted of 1024 characters (including gaps inserted for alignment). The armored dinoflagellate *Prorocentrum cordadum* (AF260379) was used as the outgroup taxon.

Phylogenetic trees for both datasets were independently constructed using maximum likelihood (ML) analyses and Bayesian inference. ML analyses were performed using PhyML ver. 3.1 [32]. The starting tree was generated using BIONJ with optimization of the topology, branch lengths, and selected rate parameters. Six different substitution rates were selected. Bootstrap analyses for both datasets were carried out using ML with 1000 replicates to evaluate the statistical reliability. Bayesian inference analyses were conducted on both datasets using the MrBayes program version 3.2 [33]. The evolutionary model used in the Bayesian inference analyses was the TVM+I+G model in the ITS region and the GTR+I+G model for the LSU region with a gamma-distributed rate variation across sites. Five Markov chain Monte Carlo (MCMC) chains were run for 10 million generations, sampling every 100 generations.

## 3. Results

### 3.1. Morphology of the Resting Cysts of Scrippsiella precaria

Cysts of *Scrippsiella precaria* were observed in sediment samples (Figure 1a,b), as well as in clonal cultures (Figure 1c–i). Cysts were yellowish or grayish, with granular cell contents (Figure 1). The cyst bodies were spherical to ovoid and the cyst diameters ranged from 18.8 to 25.6 µm (mean = 22.3 µm, $n = 30$). One or two red accumulation bodies were located toward the apical or central part of the cyst bodies (Figure 1a,c–e). The archeopyle was mesoepicystal, representing the loss of 2–4′ and 1–3a paraplates (Figure 1b). The cysts were covered with numerous narrow spines ranging in length from 0.7 to 6.6 µm (mean = 3.9 µm, $n = 30$), and the EDS indicated that the spines were composed of calcium carbonate ($CaCo_3$) (Figure 1i). The SEM observations revealed that the calcareous spines that emerged directly from the cyst bodies were pentagonal and became narrower toward the apex (Figure 1f–h). However, in the clonal culture, cysts without spiny processes were occasionally observed (Figure 2a,b). Time-lapse sequences obtained using a live-cell observation system showed the formation and changes in the length of the spiny processes of the cyst (Figure 2). Resting cysts had a rounded shape without spines until 2 days of

incubation, whereas very short spines emerged from the cyst bodies on day 3, where the spine lengths subsequently increased until day 8 of incubation (Figure 2).

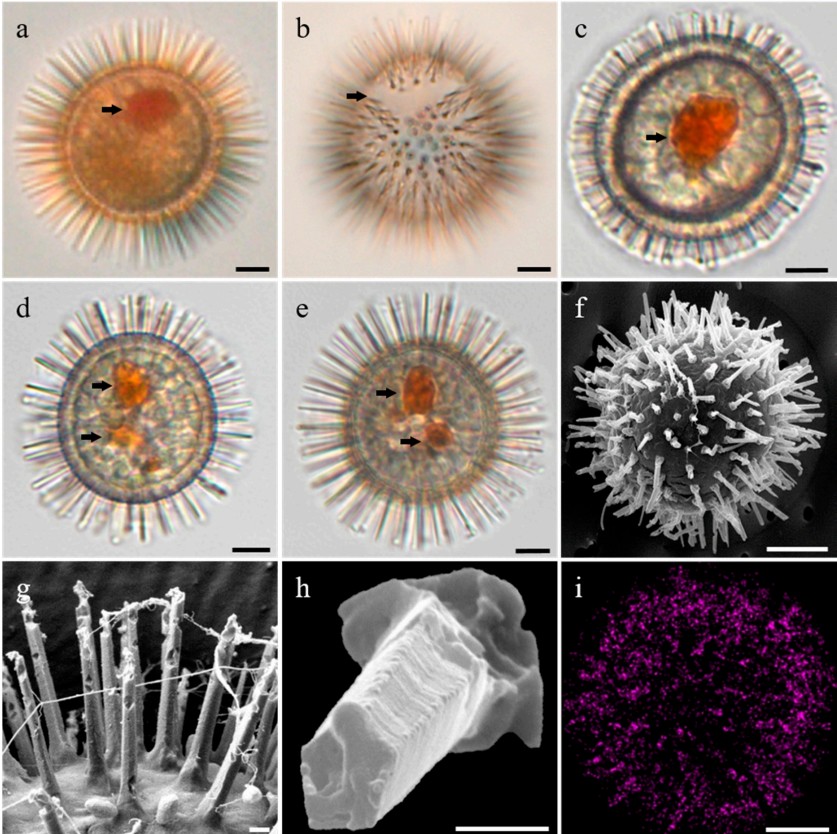

**Figure 1.** Resting cysts of *Scrippsiella precaria* from the sediment and cultures: (**a**) resting cyst from the sediment showing a red pigment body (arrow), (**b**) empty cyst showing the archeopyle (arrow), (**c–e**) resting cyst from the culture showing one or two red pigment bodies, (**f**) scanning electron microscopy (SEM) micrograph of a resting cyst, (**g,h**) details of calcareous spines, and (**i**) energy dispersive spectroscopy (EDS) result showing the detection of calcium carbonate ($CaCo_3$) on the surface of a resting cyst. Scale bars: 5 μm.

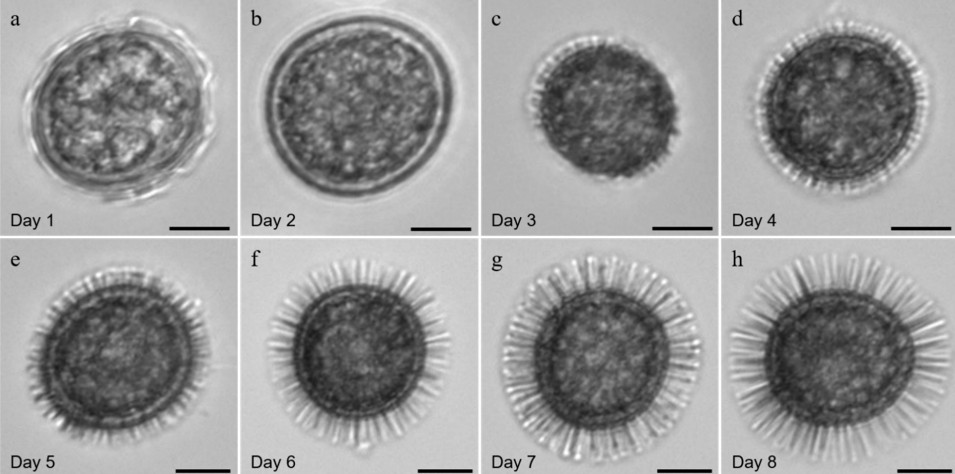

**Figure 2.** Time series of the formation and changes in the length of calcareous spiny processes of a round *Scrippsiella precaria* cyst observed in culture LMBE-C28: (**a,b**) round form of the resting cyst, (**c–g**) showing an increase in the length of the spiny processes over time, and (**h**) complete formation of a cyst. Scale bars: 10 μm.

*3.2. Morphology of Vegetative Cells of Scrippsiella precaria*

There were no significant differences in the morphology between motile cells germinated from resting cysts and those collected from water samples. Motile cells of *S. precaria* were oval and slightly dorso-ventrally compressed (Figures 3 and 4), and were usually solitary and rarely found in pairs (Figure 3e). Cells were yellowish or grayish and filled with pale white and grayish granules (Figure 3a–e), and were small, ranging from 20.4–28.5 µm in length (mean = 24.7 µm, *n* = 36) and 18.1–24.4 µm in width (mean = 21.5 µm, *n* = 36). The epitheca of the cell was conical and had a rounded apex, the cingulum was wide and the hypotheca was hemispherical (Figure 3a,c–e and Figure 4a,b). The left half of the hypotheca was sometimes prominent in the ventral view (Figure 3a,c). The epitheca was slightly longer than the hypotheca, occasionally showing a red pigment in the epitheca (Figure 3a,c–e and Figure 4a–d). A potential red eyespot was visible in the sulcal area (Figure 3c,d). In the ventral view, the spherical nucleus was visible and located in the anterior part of the cell (Figure 3f). The chloroplast was distributed toward the marginal regions of the cell but its shape was unclear (Figure 3f).

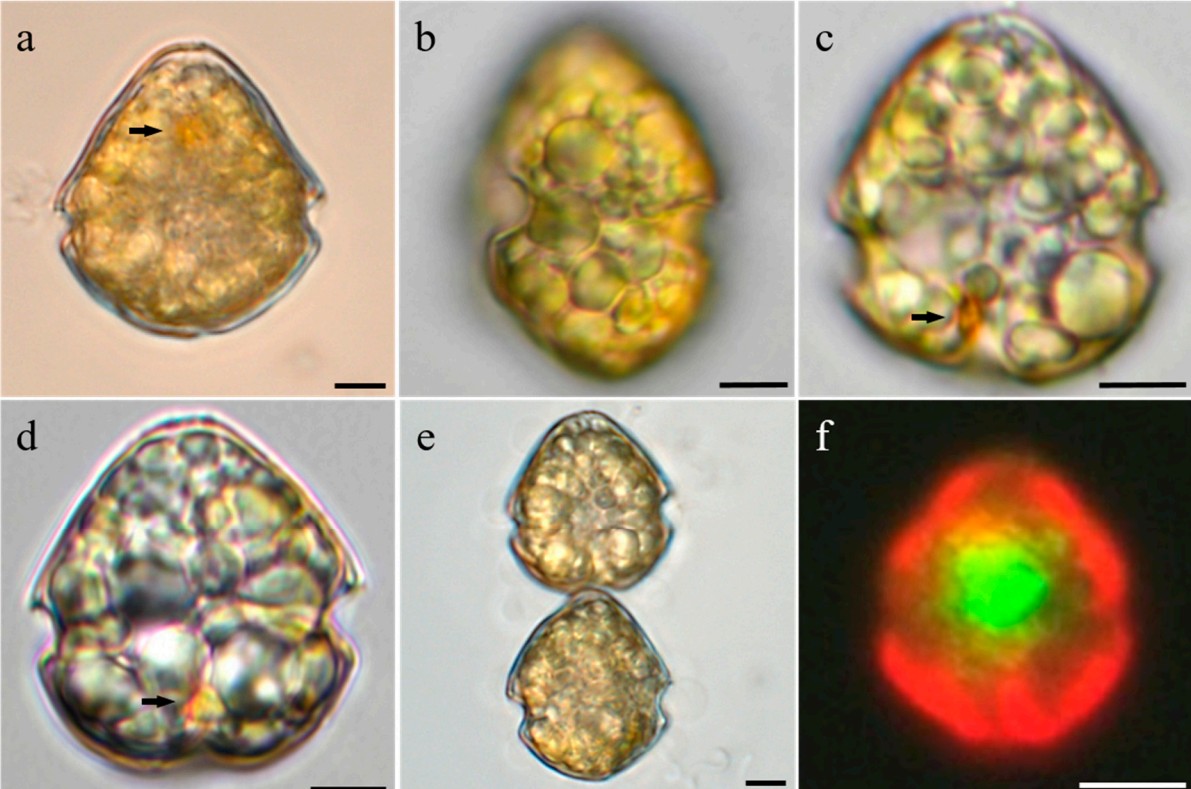

**Figure 3.** Light and fluorescence micrographs of *Scrippsiella precaria* (strains LMBE-C28 and LIMS-PS-2799): (**a**) ventral view showing a red pigment body in the epitheca, (**b**) left lateral view, (**c**) ventral view showing a red pigment body in the hypotheca, (**d**) dorsal view showing a red pigment body in the hypotheca, (**e**) two-celled chain, and (**f**) ventral view showing the position of the nucleus (green) and chloroplast (red). Scale bars: 5 µm.

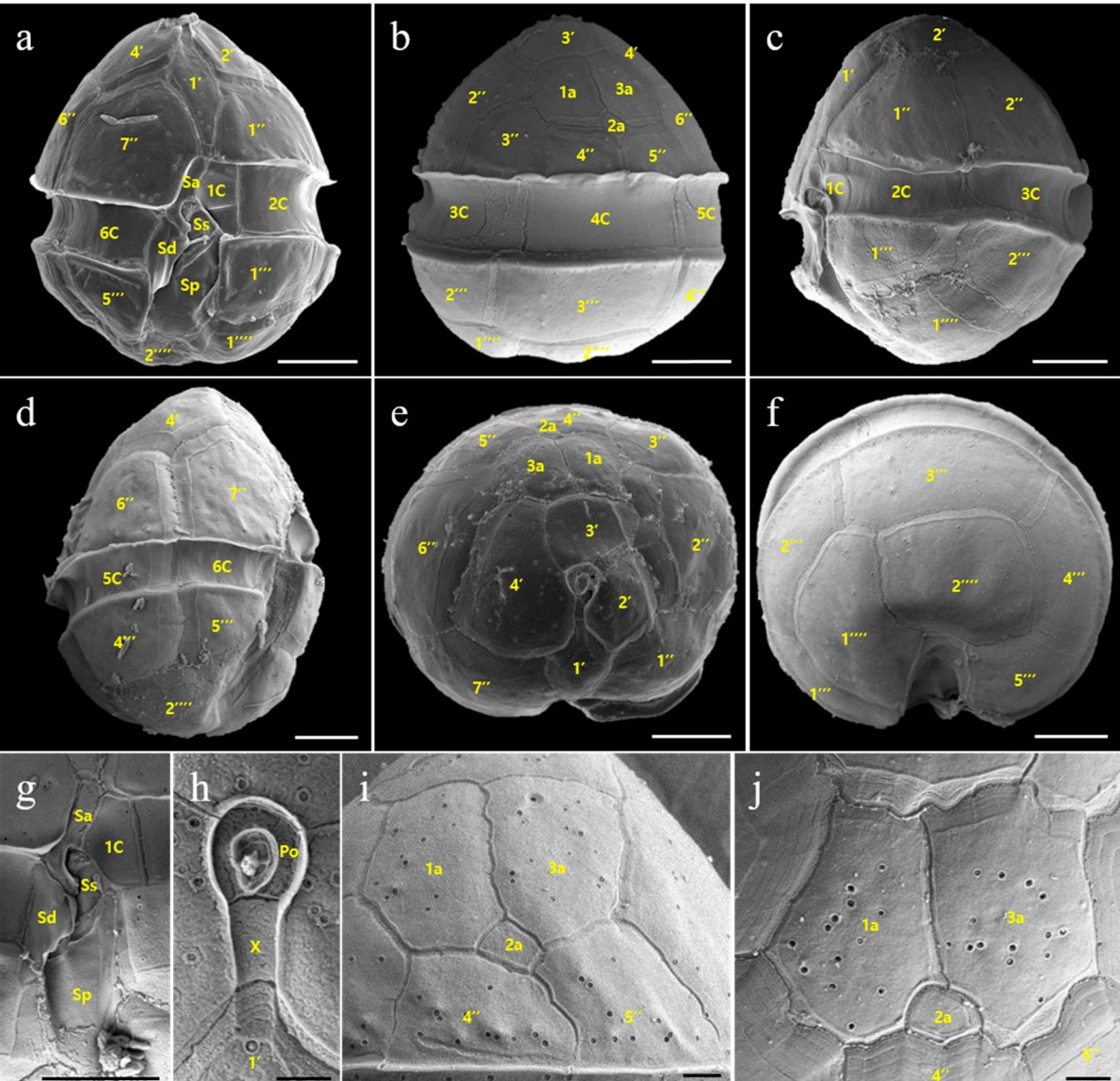

**Figure 4.** Scanning electron micrographs of *Scrippsiella precaria* (strain LIMS-PS-2799): (**a**) ventral view, (**b**) dorsal view, (**c**) left lateral view, (**d**) right lateral view, (**e**) apical view showing the apical pore (Po) and three intercalary plates, (**f**) antapical view showing the hypothecal plate pattern, (**g**) sulcal plates, (**h**) detail of Po, (**i**) detail of the 5″ plate that contacted the 2a plate, and (**j**) detail of the 5″ plate that did not contact the 2a plate. Scale bars: 5 μm (**a**–**g**) and 1 μm (**h**–**j**).

The thecal plate pattern of cells based on the SEM observations is shown in Figure 5. The cells had thin plates arranged in the Kofoidian plate formula of Po, X, 4′, 3a, 7″, 6C, 4S, 5‴, 2⁗ (Figures 4 and 5). Numerous small pores (mean diameter = 0.13 μm, *n* = 30) were randomly scattered on the thecal surface. The apical pore complex (APC), including an elongated pentagonal canal plate (X), was keyhole-shaped, comprising a polygonal pore plate (Po) surrounded by a rim and a comma-shaped pore surrounded by a rim in the middle of the APC (Figure 4h). The X plate contacted plate 1′. Plate 1′ was surrounded by plates 2′, 4′, 1″, and 7″, and was narrow and irregularly rhombic, with truncate anterior and posterior ends (Figure 4a,e). Plate 2′ was pentagonal, whereas plates 3′ and 4′ were hexagonal and heptagonal, respectively (Figure 4e). Plate 4′ was larger than plates 2′ and 3′. There were three anterior intercalary plates in the dorsal part of the epitheca and the plates contacted each other (Figure 4b,e,i,j and Figure 6). Plates 1a and 3a were hexagonal and similar in size, while plate 2a was much smaller, rhombic but occasionally triangular or rectangular, and either contacted plate 5″ or not (Figure 4b,i,j and Figure 6). Plates

3″, 4″, and 5″ were much smaller than plates 1″, 2″, 6″, and 7″ due to the presence of intercalary plates (Figure 4b,e). Plates 4″ and 5″ contacted the intercalary plates. The cingulum, comprising six plates, was wide and deeply excavated and descended to about its own width (Figure 4b,i). Among the cingulum plates, the first cingulum plate (1C) was the smallest, while the other six plates were similar in size (Figures 4a–d and 5). There were four sulcal plates. The anterior sulcal plate (Sa) was covered by the 7″ plate, contacted 1C and the right sulcal plate (Sd), and had a sickle-like extension. Sd was narrow, with a nib-shaped posterior end, and was wing-shaped on its left side (Figure 4a,g). The left sulcal plate (Ss) was covered by Sa and Sd (Figure 4a,g). The posterior sulcal plate (Sp) was wide and long and extended into the hypotheca without reaching the antapex (Figure 4a).

**Figure 5.** Schematic drawings of thecal plate patterns of *Scrippsiella precaria.* Arrowheads indicate the plate overlap pattern.

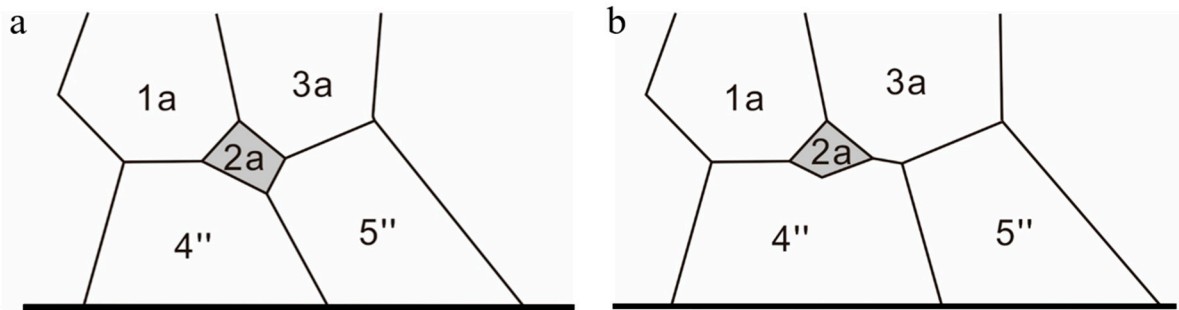

**Figure 6.** Schematic drawings of the intercalary plates and two location types of the fifth precingular plate (5″) of *Scrippsiella precaria*: (**a**) 5″ plate that contacted the 2a plate and (**b**) 5″ plate that did not contact the 2a plate.

In the postcingular series, plate 1‴, which contacted plates Sp, 1C, 2C, 1⁗, and 2⁗, was tetragonal and wider than the other four plates (Figure 4a). Plate 3‴ was the largest postcingular plate and contacted the antapical plates (1⁗ and 2⁗). Plate 2‴ was the second-largest plate and only contacted plate 1⁗. Plates 4‴ and 5‴ contacted plate 2⁗, whereas plate 5‴ also contacted plates Sp and Sd. The antapical plates were pentagonal and similar in size and contacted plate Sp.

The plate overlap patterns of the epithecal, cingular, and hypothecal plate series followed two general gradients: from dorsal to ventral and from cingulum to the two poles (Figure 5). The fourth precingular (4″) and third postcingular (3‴) plates and the fourth cingular (4C) plate were identified as keystone plates that overlapped all adjacent plates. In the sulcal plate series, the Sp plate was overlapped by the cingulum and hypothecal plates.

### 3.3. Molecular Phylogeny of Scrippsiella precaria

The inferred phylogenies from the ML and Bayesian inference (BI) analysis based on ITS region (ITS1, ITS2, and 5.8S rDNA) and LSU rDNA sequences are shown in Figures 7 and 8. The sequence of one strain of *Scrippsiella precaria* (LIMS-PS-2799) for LSU rDNA, and the sequences of two strains of *S. precaria* (LIMS-PS-2799 and LMBE-C28) for the ITS region, with other related species available from GenBank, were used for the phylogenetic analysis.

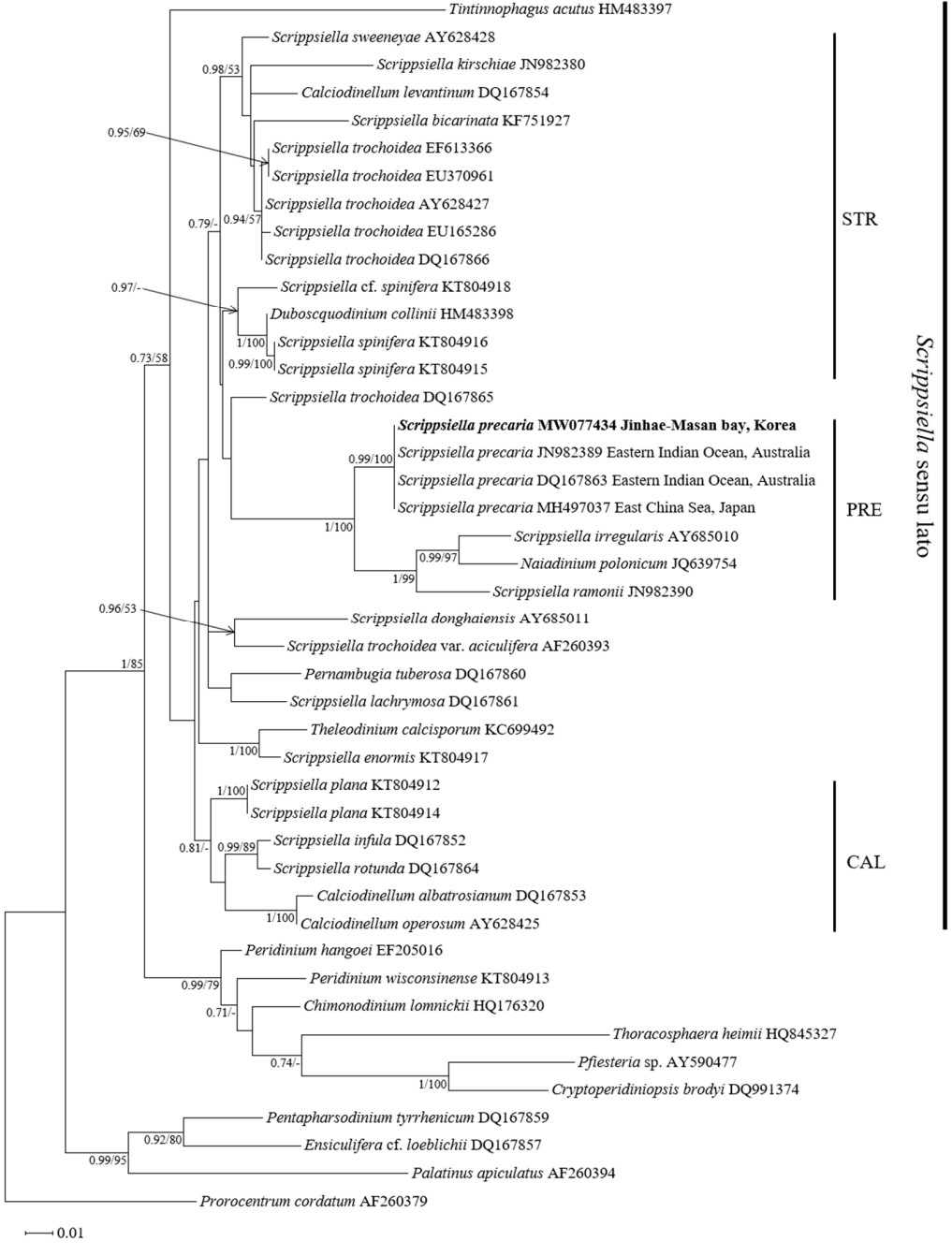

**Figure 7.** Phylogenetic positions of Korean isolates of *Scrippsiella precaria* that were inferred from the LSU rDNA sequences based on maximum likelihood (ML). The numbers on each node are the Bayesian posterior probability (PP) followed by the bootstrap values (%). Only bootstrap values above 50% and PP above 0.7 are shown. CAL: clade of *Calciodinellum* and its relatives, STR: clade of *S. trochoidea* and its relatives, PRE: clade of *S. precaria* and its relatives. These clades were based on the results of Gottschling et al. [18] and Luo et al. [15]. Scale bar: 0.01 nucleotide substitutions per site.

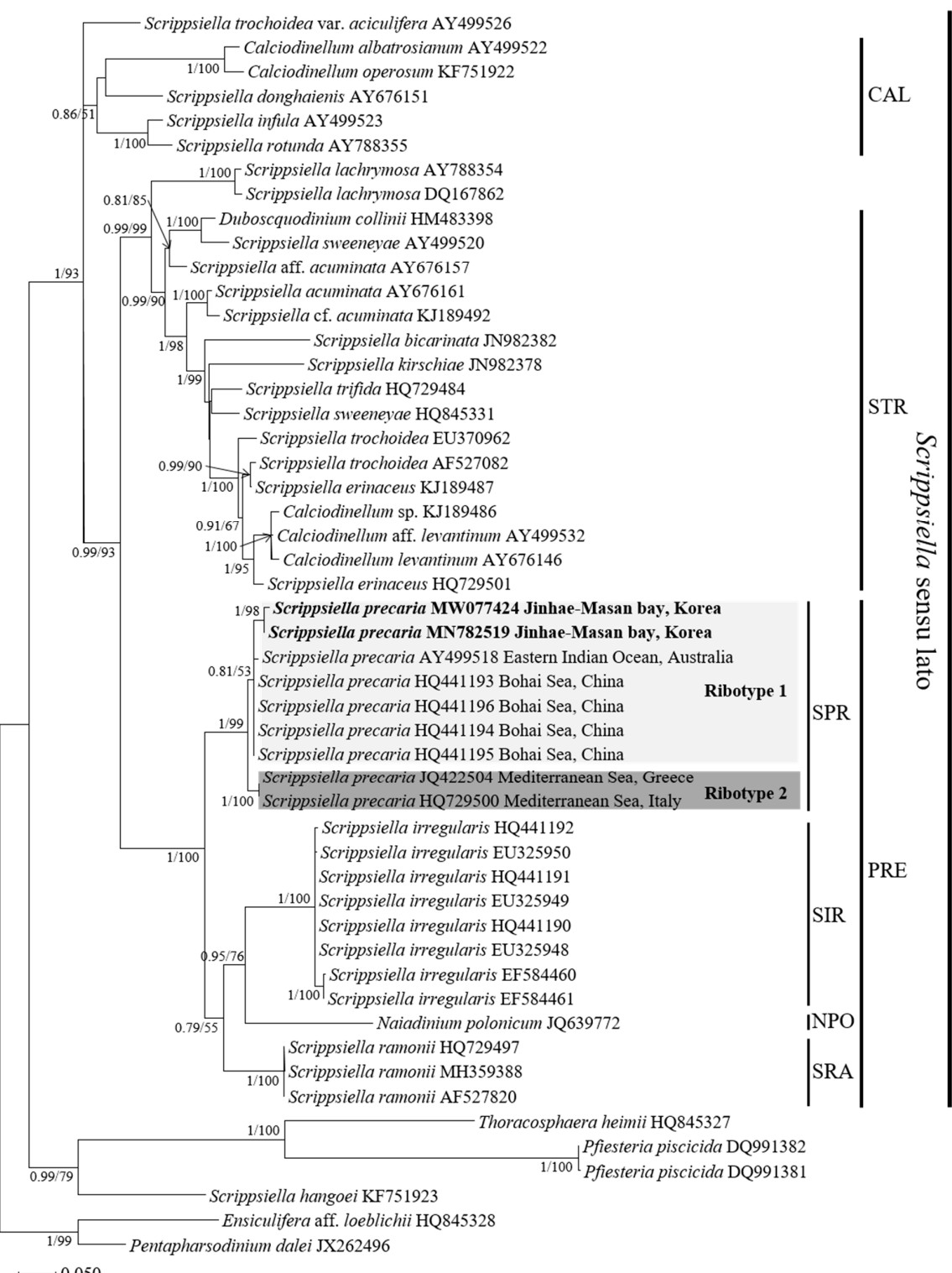

**Figure 8.** Phylogenetic positions of Korean isolates of *Scrippsiella precaria* that were inferred from the ITS and 5.8S sequences based on ML. *Ensiculifera* aff. *loeblichii* and *Pentapharsodinium dalei* were used as outgroup taxa. The numbers on each node are the Bayesian PP followed by the bootstrap values (%). Only bootstrap values above 50% and PP above 0.7 are shown. STR: clade of *Scrippsiella trochoidea* species and its relatives, PRE: clade of *Scrippsiella precaria* and its relatives; CAL: clade of *Calciodinellum* and its relatives; SPR: subclade of PRE, consisting of *S. precaria*; SIR: subclade of PRE, consisting of *S. irregularis*; NPO: subclade of PRE, consisting of *Naiadinium polonicum*; SRA: subclade of PRE, consisting of *S. ramonii*. These clades were based on the results of Gottschling et al. [18] and Luo et al. [15]. Scale bar: 0.05 nucleotide substitutions per site.

ML and BI based on ITS region and LSU rDNA generated similar phylogenetic trees that differed by only a few topological features. The molecular tree based on LSU rDNA formed a clade (*Scrippsiella* sensu lato) that mainly consisted of *Scrippsiella* species and those of genera *Calciodinellum*, *Duboscquodinium*, *Naiadinium*, *Thelodinium*, and *Tintinnophagus* (Figure 7); however, this clade had weak support (posterior probability/ML bootstrap = 0.73/58). Three subclades, namely, CAL (*Calciodinellum* and related species), STR (*Scrippsiella trochoidea* and related species), and PRE (*S. precaria* and related species), were identified (Figure 7), and the Korean strain of *S. precaria* was nested within the PRE clade and shared an identical sequence with the Australian (DQ167863 and JN982389) and Japanese (MH497037) strains of *S. precaria* (Figure 7).

The molecular tree based on the ITS region had a similar topology to the LSU rDNA gene tree; however, the clade of *Scrippsiella* sensu lato received strong support (1/93) (Figure 8) and comprised three clades (CAL, STR, and PRE). Korean strains of *S. precaria* were nested within the PRE clade, which consisted of four subclades: the SPR clade containing *S. precaria*, the SIR clade containing *S. irregularis*, the NPO clade consisting of *Naiadinium polonicum*, and the SRA clade composed of *S. ramonii* (Figure 8). In the SPR clade, two ribotypes were identified among the Chinese, Korean, Australian, Italian, and Greek strains of *S. precaria*. Korean strains of *S. precaria* were nested within ribotype 1 and grouped with Chinese and Australian strains (0.81/53), whereas ribotype 2 was composed of Italian and Greek strains (1/100) of *S. precaria*.

## 4. Discussion

### 4.1. Morphological Features of Resting Cysts of Scrippsiella precaria

According to Gu et al. [13], the presence of a red body and the shape of the archeopyle in resting cysts of *Scripsiella* species, including *S. precaria,* are potentially useful taxonomic characters for the identification of the species because these characteristics are constant among strains and from one generation to the next generation of cysts. Montresor and Zingone [23] also described resting cysts of *S. precaria* with a red body. However, resting cysts of *S. precaria* in this study had one or two prominent red bodies. Variability in the number of red bodies has been observed in *Scrippsiella spinifera* [15]. This indicates that the number of red bodies should not be used as a taxonomic character to distinguish between vegetative cells of *Scrippsiella* species and other related species. By contrast, the shape of the archeopyle appears to be a stable character because *Scrippsiella* species, including *S. precaria,* usually have a cap-shaped operculum that includes apical (2′–4′) and/or intercalary (1a–3a) paraplates (e.g., [4]). Such an archeopyle was found in *S. precaria* and has also been reported in *Scrippsiella regalis*, *S. trochoidea*, *Scrippsiella rotuda*, and *S. donghaiensis* [12,13].

Cysts of *Scrippsiella* species are usually surrounded by calcareous ornaments [2,3,34,35], whereas some species can produce noncalcareous cysts [15,22]. Previous studies on cyst–theca relationships in *S. precaria* revealed that resting cysts are characterized by calcareous spines with capitate or pointed ends [20,23,24]. Calcareous spines of resting cysts collected from Korean sediments and those formed in clonal cultures also had pointed spines. However, the capitate ends recorded by Monstresor and Zingone [23,24] were not observed in this study, and the spine ends of resting cysts of *S. precaria* from Chinese and Japanese coastal areas were pointed as well (e.g., [20,26]). This indicates that *S. precaria* has two types of spines. According to Montresor et al. [19], the morphology of calcareous ornamentations can be influenced by environmental conditions, such as macro- and micro-nutrient concentrations, pH, and light availability. We were not able to address this issue based on the data obtained in our study. Differences in spine type may be related to differences in environmental conditions based on the geographic origins of *S. precaria* because strains of *S. precaria* from Asian coastal areas, such as the Bohai Sea of Chinese and Korean coast, have identical ITS and LSU rDNA sequences (Figures 7 and 8).

Several studies reported resting cysts of *Scrippsiella* species without calcareous ornaments in culture, despite the observation of spiny cysts in the field (e.g., [13,24,36,37]). Wang et al. [36] and Shin et al. [37] reported two morphotypes of *S. trochoidea* cysts from

surface sediments and cultures: a typical type with short calcareous spines and a transparent type without calcareous spines (naked-type cyst). Montresor and Marino [24] observed the formation of spines from the rounded form of resting cysts of *S. precaria* that were identical to our observations. These observations indicate that true resting cysts of calcareous *Scrippsiella* species can have a rounded form. Shin et al. [37] also observed that in acidic environments, the lengths of calcareous spines were shorter than those observed for typical cysts and the shape of the spines was abnormal, indicating that the length and shape of calcareous spines of *Scrippsiella* species are plastic and therefore unreliable characteristics for classification [13].

According to Shin et al. [37], the formation of calcareous spines can help to protect the cyst body from aerobic and anaerobic decay or mechanical damage during ingestion and digestion by predators. If so, the rapid formation of calcareous spines may ensure cyst survival. In this study, it took 8 days for the complete formation of calcareous spines of *S. precaria*. This was the first record of the time required to form calcareous spines in *Scrippsiella* species; however, because the relationship between cyst survival and the formation time of calcareous spines remains unknown, further studies are needed.

### 4.2. Morphological Comparisons of Korean Isolates of Scrippsiella precaria with Previously Described Species

According to Montresor and Zingone [23] and Gu et al. [20], *Scrippsiella precaria* specimens from the Gulf of Naples and the Bohai Sea have the plate formula of Po, X, 4′, 3a, 7″, 6C, 5S, 5‴, 2⁗. They described a median sulcal plate (Sm) in the sulcal region, whereas no Sm plate was observed in the Korean isolates. This was possibly because the Sm was almost hidden by the wing of the right sulcal plate [23]. *S. precaria* described from previous studies was characterized as being slightly compressed dorso-ventrally, with the presence of a red pigment body and spherical nucleus, where these morphological features are consistent with those observed in Korean isolates of *S. precaria*. A red pigment body in the lower sulcal area was observed in *S. precaria* specimens collected from the Gulf of Naples [23,24], which is consistent with our findings in Korean isolates. However, reported cell sizes of isolates of *S. precaria* differed between studies: isolates from the Gulf of Naples and the Bohai Sea, China, have similar sizes (15–25 μm in length and 13.5–20 μm in width) [20,23], but isolates from Korean and Japanese coastal waters are slightly larger (e.g., [25,26]).

*Scrippsiella* species usually have three intercalary plates, with plates 1a and 3a separated by plate 2a (e.g., [13,15,18,19,22]). However, in some *Scrippsiella* species, such as *S. irregularis*, *S. ramonii*, and *S. precaria*, the plates 1a and 3a contact each other, and thus the 2a plate contacts not only the 1a and 3a plates, but also two precingular plates (4″ and 5″) [23,34,38]. Gu et al. [20] described the 2a plate of *S. precaria* collected from the Bohai Sea of China as touching the 4″ and 5″ precingular plates. However, Korean isolates of *S. precaria* had two types of the 5″ plate: one type that contacted the 2a plate and the other that did not. It is possible that Gu et al. [20] did not observe a specimen in which the 5″ plate did not contact the 2a plate because in most *S. precaria* cells, the 2a plate contacts the 4″ and 5″ plates. According to Attaran-Fariman and Bloch [38] and Gu et al. [20], differences in the size of the 2a plate and the shape of the hypotheca are key characteristics that can be used to separate *S. precaria* from *S. irregularis* and *S. ramonii*. Consequently, the smaller size of the 2a plate, the shape of the hypotheca, and the locations of the 5″ plate may be distinct features that distinguish *S. precaria* from other *Scrippsiella* species.

The plate overlap pattern is considered to be conserved at higher taxonomic levels [39]. The epithecal plate overlap pattern of *S. precaria* is the same as that found in most peridinioid dinophytes [3,40], with the fourth precingular plate forming the keystone. The 4C plate is the keystone plate of the cingular series of *S. precaria*, identical to *S. acuminata* (heterotypic synonym of *S. trochoidea*) [3].

*4.3. Phylogenetic Position of the Korean Isolates of Scrippsiella precaria*

The molecular phylogeny of *Scrippsiella precaria* based on the ITS and LSU sequences in the present study was consistent with those reported in previous studies (e.g., [15,18,20]); in a *Scrippsiella* sensu lato clade consisting of mainly *Scrippsiella* species, *S. precaria* was nested within the PRE clade. Gu et al. [20] found that only one *S. precaria* isolate from the Bohai Sea of China was a member of the PRE clade; however, Luo et al. [15] found that an isolate from Australia was grouped with the Chinese isolate in the PRE clade. This result agrees with the results of the current study. In addition, the PRE clade was divided into four subclades (SPR, SIR, NPO, and SRA) in our phylogeny based on the ITS region, and the Korean isolates were nested within the SPR clade (Figure 8). The SPR clade comprised two ribotypes of *S. precaria*: ribotype 1 consisting of isolates from Korea, China, and Australia, and ribotype 2 consisting of isolates from Italy and Greece. According to Shin et al. [41,42] and Hallegraeff [43], the dispersal of harmful microalgae, such as *Alexandrium* species in Asian coastal areas and Australia, is related to ocean currents and ballast waters via international shipping, and because of this, the harmful species from Jinhae-Masan Bay, Korea, are phylogenetically grouped with those from China and Australia. This suggests that the lineages between isolates of the ribotype 1 could also be due to dispersal. In the SPR clade, Korean isolates are characterized by cysts with pointed spines, as was also reported for the Chinese and Japanese strains [20,26], while the Italian cysts are spiny and mostly have a capitate end [23]. In addition, the 5″ plate either contacts or does not contact the 2a plate, which may be a distinguishing feature of ribotype 1, although 5″ plates that do not contact the 2a plate have not been found in the isolates from China and Australia. Consequently, further studies are needed to clarify the morphological differences between the ribotypes of *S. precaria*.

**Author Contributions:** Data curation, formal analysis, writing—original draft, and writing—review and editing, H.J.K., Z.L. and H.H.S.; funding acquisition, H.H.S.; data curation and investigation, N.S.K.; writing—review and editing, H.G., D.K., M.H.S., S.D.L., S.M.Y. and S.-J.O. All authors have read and agreed to the published version of the manuscript.

**Funding:** This work was supported by grants from the Korea Institute of Ocean Science & Technology (KIOST) (PE99821), the National Marine Biodiversity Institute of Korea (MABIK) (2021M01100), and the Nakdonggang National Institute of Biological Resources (NNIBR) (202101103) projects, and the Korea Research Institute of Bioscience and Biotechnology (KRIBB) Research Initiative Program.

**Institutional Review Board Statement:** Not applicable.

**Informed Consent Statement:** Not applicable.

**Conflicts of Interest:** The authors declare no conflict of interest.

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
