# Peer review of "Morphology and Phylogeny of Scrippsiella precaria Montresor & Zingone (Thoracosphaerales, Dinophyceae) from Korean Coastal Waters"

_jmse, doi:10.3390/jmse9020154_

Round 1

Reviewer 1 Report

Review of the Manuscript JMSE ID-1072818

Title: Morphology and phylogeny of Scrippsiella precaria Montresor&Zingone (Thoracosphaerales, Dinophyceae) from Korean coastal waters

Authors: Hyun Jung Kim, Zhun Li, Nam Seon Kang, Haifeng Gu, Daekyung Kim, Min Ho Seo, Sang Deuk Lee, Suk, Min Yun, Seok-Jin Oh and Hyeon Ho Shin

Date: January, 10th 2021

In this study the authors give detailed morphological characterization of dynoflagelate species Scrippsiella precaria. The analyzed cells were germinated from the resting cysts isolated from sediment off the Korean coastal waters. Authors have performed a thorough phylogenetical analysis of the isolated specimens and comparison of its ribotype to the ones isolated from Korea, China, Australia, and Italy and Greece. The sequencing results indicate relationship between morphological differences and geographic origins of S. precaria strains from Asian coastal areas. The manuscript is very systematic and well written, the study is well designed and discussion is scientifically sound and complete. All the relevant literature is taken into consideration. Overall, the Manuscript presents valuable results that contribute significantly to the global understanding of marine plankton taxonomy and ecology. I recommend its publication in its present form.

Minor comments:

Since results were partly presented in the frame of a conference (https://sciwatch.kiost.ac.kr/handle/2020.kiost/22238) I recommend a modification of the title to indicate the enrichment of the study presented in the full-length research article.

Line 115 (and throughout whole text): please write f/2-Si with capital “S” since Si is a chemical symbol

Line 116 (and throughout whole text): please write 20°C without space between value and degrees.

The whole text in the Manuscript should be vertically aligned to the same margins on left and right side.

Author Response

#Reviewer 1

In this study the authors give detailed morphological characterization of dynoflagelate species Scrippsiella precaria. The analyzed cells were germinated from the resting cysts isolated from sediment off the Korean coastal waters. Authors have performed a thorough phylogenetical analysis of the isolated specimens and comparison of its ribotype to the ones isolated from Korea, China, Australia, and Italy and Greece. The sequencing results indicate relationship between morphological differences and geographic origins of S. precaria strains from Asian coastal areas.

The manuscript is very systematic and well written, the study is well designed and discussion is scientifically sound and complete. All the relevant literature is taken into consideration. Overall, the Manuscript presents valuable results that contribute significantly to the global understanding of marine plankton taxonomy and ecology. I recommend its publication in its present form.

Response – Thank you for your careful evaluation. Please see the revised manuscript for corrections.

Minor comments:

Comment 1: Since results were partly presented in the frame of a conference (https://sciwatch.kiost.ac.kr/handle/2020.kiost/22238) I recommend a modification of the title to indicate the enrichment of the study presented in the full-length research article.

Response – We think the current title is suitable for this study.

Comment 2: Line 115 (and throughout whole text): please write f/2-Si with capital “S” since Si is a chemical symbol

Response – Corrected.

Comment 3: Line 116 (and throughout whole text): please write 20°C without space between value and degrees.

Response - Corrected

Comment 4: The whole text in the Manuscript should be vertically aligned to the same margins on left and right side.

Response – The whole text in the manuscript was automatically aligned after submission. I think the alignment will be changed by publisher.

Reviewer 2 Report

  1. Please reconstruct the whole manuscript according to the guidelines of the journal of the Marine Science and Engineering (Lines 89-91,158-163, 180-190). 
  2. If possible, please improve the quality of the figures 1a-d and Figure 2.
  3. Improve the clarity of the sentence Lines 66-69
  4. The references should be up-to-date. Among 42 references, there are only three studies from 2018 and two from 2019.
  5. Line 442: The financial disclosure should not be under subheading:Acknowledgements.
  6. The authors should mention, by whom the schematic drawings have been conducted. When the drawings were not performed by a co-author, please acknowledge.

Author Response

#Reviewer 2 (Please see the rivised manuscript for corrections)

Comment 1: Please reconstruct the whole manuscript according to the guidelines of the journal of the Marine Science and Engineering (Lines 89-91,158-163, 180-190). 

Response – We have checked and revised the manuscript.

Comment 2: If possible, please improve the quality of the figures 1a-d and Figure 2.

Response – Fig. 1 was made, with high resolution, but we will provide other figures when publisher requests.

Comment 3: Improve the clarity of the sentence Lines 66-69

Response – We have corrected the sentence. Please see the revised manuscript.

Comment 4: The references should be up-to-date. Among 42 references, there are only three studies from 2018 and two from 2019.

Response – Although Scrippsiella precaria is cosmopolitan species, the morpho-molecular phylogeny of the species has been rarely reported. The references in the manuscript are all of recent studies on S. precaria.

Comment 5: Line 442: The financial disclosure should not be under subheading: Acknowledgements.

Response - The whole text in the manuscript was automatically aligned after submission. I think the alignment will be changed by publisher.

Comment 6: The authors should mention, by whom the schematic drawings have been conducted. When the drawings were not performed by a co-author, please acknowledge

Response – We made the drawings and figures in the manuscript.

Reviewer 3 Report

The manuscript is very well written and provide sufficient valuable information. Minor correction are desirable to improve overall quality of publication as follows:

  1. Introduction: this section contains excessive information about the whole dinoflagellate genus Scrippsiella that took three paragraphs whereas the study focusses on Scrippsiella precaria only. It should be better to shorten first part of introduction and extend the part devoted to the S. precariat.
  2. Results of the thecal plate morphology are very descriptive and require some objective measurement that allow to perform a comparison of different parts of this object.
  3. Results and Discussion section contains a lot of data about Scrippsiella precariat and comparisons with that species from different regions. It would be better to summarize main findings into short Conclusion section.

Author Response

#Reviewer 3

Comment 1: The manuscript is very well written and provide sufficient valuable information. Minor corrections are desirable to improve overall quality of publication as follows:

Introduction: this section contains excessive information about the whole dinoflagellate genus Scrippsiella that took three paragraphs whereas the study focusses on Scrippsiella precaria only. It should be better to shorten first part of introduction and extend the part devoted to the S. precariat.

Response – We do not agree with this suggestion. The introduction in the manuscript includes the information on Scrippsiella sensu lato clade, because Scrippsiella precaria is nested within the clade. We think that the information should be clarified in the Introduction section

Comment 2: Results of the thecal plate morphology are very descriptive and require some objective measurement that allow to perform a comparison of different parts of this object.

Response – We think all morphological descriptions were provided for identification of Scrippsiella precaria.

Comment 3: Results and Discussion section contains a lot of data about Scrippsiella precariat and comparisons with that species from different regions. It would be better to summarize main findings into short Conclusion section.

Response – Thank you for your suggestion, but we think abstract section contains enough information.

Reviewer 4 Report

This ms by Kim et al. is very interesting with noteworthy results, adding new knowledge to the morphology of different life cycle stages of Scrippsiella precaria and its phylogeny.

I recommend the publication on JMSE with minor revisions.

In fact, I noted many inaccuracies throughout the text, so, please see my suggestions in the attached file "comments"

Author Response

#Reviewer 4

This ms presents the results of an interesting study concerning the morphology of active and resting stages of the dinoflagellate Scrippsiella precaria and the phylogeny of this species from Korean coastal waters.

The Authors make a very interesting comparison with the results of similar studies in the Atlantic and the Mediterranean contributing in a very extensive way to a better knowledge of the life cycle stages of the species and its phylogeny.

All the sections of the ms are well developed and presented, the methods adopted are very effective and the results are presented and explained in an exhaustive way. The cited literature is complete also concerning the genetic methods.

I only have one general remark concerning the language style that in many cases is complicated by many repetitions which make the reading quite difficult and two remarks about a couple of sections of the ms:

Response – Thank you for your careful evaluation. Please see revised manuscript for correction.

Comment 1: 1. the paragraph 2.1 “Sample collection and culture of the Methods” is quite confusing, also concerning the english and in my opinion should be revised (please see my comments). The Authors should pay attention to the use of singular and plural, in particular starting from line 113

Response – Thank you. English use has been checked by native speakers, again. Please see the revised manuscript.

Comment 2: 2. the paragraph 4.3 “Phylogenetic position of Korean isolates of Scippsiella precaria” of the Discussion deserves more attention by the Authors, due to the interesting results showed.

Response – Thank you for your suggestion, but this study provides only morpho-molecular characterization of Scrippsiella precaria from Korean coastal areas. In addition, as we have no data on S. precaria collected from other Asian coastal areas (excluding China), we think further studies on phylogenetic relationships are needed.

Specific comments and suggested revisions

Abstract

Comment 3: line 21 ...of phytoplankton and, despite their cysts are….. worldwide, the diversity of Scrippsiella in…….

Response - Corrected

Comment 4: l 27 Rounded resting cysts were found in culture…

Response - Corrected

Comment 5: l 30 …previously described cells of S. precaria….

Response - Corrected

Comment 6: l 34 two ribotypes [please delete: of S. precaria], the ribotype 1…….. Australia and the ribotype 2…

Response - Corrected

Introduction

Comment 7: l 48 in my opinion the references from 6 to 10 are not necessary here, maybe it could be more useful to cite only Elbrächter et al. 2008 (Elbrächter, M.; Gottschling, M.; Hildebrand-Habel, T.; Keupp, H.; Kohring, R.; Lewis, J.; Meier, K.S.; Montresor, M.; Streng, M.; Versteegh, G.J.; Willems, H.; Zonneveld, K.; Establishing an agenda for calcareous dinoflagellate research (Thoracosphaeraceae, Dinophyceae) including a nomenclatural synopsis of generic names. Taxon 2008, 57, 1289–1303.)

Response – Thank you. We have added the reference.

Comment 8: l 56 Calciodinellum Deflandre

Response – In first paragraph, authority was provided

Comment 9: l 57 …the issue of the species included within….

Response - Corrected

Comment 10: l 73 ...these species form a clade

Response - Corrected

Comment 11: l 80 …species together with calcareous resting cysts were found. The cultures established, based on the incubation of collected vegetative cells and germination experiments, were examined by light….

Response - Corrected

Comment 12: l 85 …and then we compare them with those previously reported in other studies.

Response - Corrected

Materials and methods

Comment 13: Maybe we need a figure of the study area here with the location of the sampling station

Response – I think we can find the information of Jinhae-Masan Bay, easily, because this bay has been studied by many researchers. So we did not show the area.

Comment 14: l 89 when the sampling was carried out?

Response – We have added the date.

Comment 15: 106 After sufficient… - please explain sufficient (also at line 118), it means a sufficient number of cells or what else?

Response – It means the increase in cell number.

Comment 16: l 111 …a station in the Jinhae-Masa Bay

Response - Corrected

Comment 17: l 113 here the use of singular/plural creates some confusion: only one cell was isolated from the plankton sample and transferred into one well or they were more?

Response – Yes, only one cell was isolated and incubated. To avoid confusion, we have checked the use of singular/plural.

Comment 18: l 114 The isolated cells were individually inoculated into the wells of a 48-well…

Response - Corrected

Comment 19: l 115 I suggest to move the manufacturer of the medium to line 103 where the medium is reported for the first time

Response - Corrected

Comment 20: l 134 …in the morphology of the resting cysts, …

Response - Corrected

Comment 21: l 135 please check the code of the strain, I don’t think it is correct

Response – The code name is correct. The code name represent the vegetative cell germinated from a resting cyst

Comment 22: l 135 …into a Petri dish…

Response - Corrected

Comment 23: l 170 please check the website url, I think it is www.dnabaser

Response - Corrected

Comment 24: l 185 …were conducted on…

Response - Corrected

Comment 25: l 200 …the spines as composed by calcium…. (CaCO3)

Response - Corrected

Comment 26: l 210 …please uniform sediment or sediments everywhere

Response - Corrected

Comment 27: l 212 please indicate the acronym in the caption: EDS (Energy Dispersive Spectroscopy)

Response – The acronym has been added.

Comment 28: l 221 …and those collected from…

Response - Corrected

Comment 29: l 226 of the cell…

Response - Corrected

Comment 30: l 227 hemispherical…

Response - Corrected

Comment 31: l 228 …the hypotheca, occasionally showing a red pigment

Response - Corrected

Comment 32: l 230 …the spherical nucleus was visible and…

Response - Corrected

Comment 33: l 234 the plate formula comprises a question mark to be deleted, or maybe it means there is difference in the sulcus plate arrangement of this study (4 sulcal plates) compared with the classical plate formula of the species described by Montresor and Zingone with 5 sulcal plates? If so, it should be explained

Response – We have deleted a question mark.

Comment 34: l 235 maybe randomly is useless

Response – We think there is no problem for the sentence.

Comment 35: l 245 I suggest to delete: In seven precingular plates.

Response – We have deleted that.

Comment 36: l 246 Plates 3’’, 4’’ and 5’’ were… l 249 (Fig. 4b, c, d)…..plate (1C) was the smallest, while the other six plates….

Response - Corrected

Comment 37: l 253 …posterior end, and a wing-shaped left side…

Response - Corrected

Comment 38: l 257 …the plates Sp, 1C, 2C, 1’’’’ and 2’’’…

Response - Corrected

Comment 39: l 258 …Plate 3’’’ was the largest among the post-cingular….

Response - Corrected

Comment 40: l 285 The phylogeny inferred from…

Response - Corrected

Comment 41: l 292 …differed by only…

Response - Corrected

Comment 42: l 293 …mainly consisting of Scrippsiella species, togeher with…

Response - Corrected

Comment 43: l 295 (Fig. 7); however…

Response - Corrected

Comment 44: l 297 …related species), were identified…

Response - Corrected

Comment 45: l 299 Australian (DQ167863….) and Japanese strains…

Response - Corrected

Comment 46: l 300 …showed a similar pattern…

Response - Corrected

Comment 47: l 301 …in the tree, consisting of the same three subclades (CAL, STR and PRE), received…(Fig. 8). Korean strains..

Response - Corrected

Comment 48: l 303 …four subclades: SPR clade consisting of S. precaria, SIR clade consisting of….

Response - Corrected

Comment 49: l 305 consisting of S. ramonii… Figs 7 and 8 report in the trees the values of the Bayesian probability and then the bootstrap, while in the captions the order is inverted

Response - Corrected

Discussion

Comment 50: l 329 characters for the identification of the species…

Response - Corrected

Comment 51: l 333 …of red bodies has been observed in…

Response - Corrected

Comment 52: l 334 indicates that this cannot be a taxonomic…

Response - Corrected

Comment 53: l 335 the cysts of Scrippsiella at species level….

Response - Corrected

Comment 54: l 336 … the shape of the archeopyle seems to be a stable character…

Response - Corrected

Comment 55: l 339 …but it is reported in S. regalis… S. rotunda…

Response - Corrected

Comment 56: l 343 by calcareous spines…

Response - Corrected

Comment 57: l 345 … the capitate ends recorded by Montresor and Zingone [22] were not…

Response - Corrected

Comment 58: l 349 types of spines…

Response - Corrected

Comment 59: l 352 …in the shape of the spines…

Response - Corrected

Comment 60: l 353 this feature may be related…

Response - Corrected

Comment 61: l 359 … morphotypes…

Response - Corrected

Comment 62: l 360 …and another one without…

Response - Corrected

Comment 63: l 362 …from the rounded form…

Response - Corrected

Comment 64: l 363 … as reported from our experiments…

Response - Corrected

Comment 65: l 364 …can have a a rounded form…

Response - Corrected

Comment 66: l 366 shorter and the shape abnormal, indicating that these characters of calcareous….

Response - Corrected

Comment 67: l 369 …to protect the cyst body…

Response - Corrected

Comment 68: l 371 …process in the formation of…

Response - Corrected

Comment 69: l 372 …of the cysts. In this study, for the first time, the complete formation of calcareous spines of S. precaria, was observed, and this process needed eight days. However, since…

Response - Corrected

Comment 70: l 375 of cyst and the time period required for the formation of calcareous….

Response - Corrected

Comment 71: l 383 …isolates was not observed, possibly because…

Response - Corrected

Comment 72: l 385 …by a slight dorso-ventral compression, the presence…

Response - Corrected

Comment 73: l386 …of Korean isolates. A red pigment….

Response - Corrected

Comment 74: l 388 …sulcal area [22, 23] and in the Korean isolates as well.

Response - Corrected

Comment 75: l 389 …previous studies; in fact, the isolates from the Gulf of Naples and Bohai Sea, China, exhibited similar sizes (15-25….) [19, 22], while those from Korean and Japanese coastal waters were slightly larger [e.g., 24, 25].

Response - Corrected

Comment 76: l 394 …with 1a and 3a separated by the 2a plate […..17, 18, 21], even though, in some species such as S. irregularis, S. ramonii and S. precaria, the plates 1a and 3a contacted each other..

Response - Corrected

Comment 77: l 397 please check this sentence because you wrote that the plate 2a contacts the 1a and 2a!

Response - Corrected

Comment 78: l 399 …2a plate touching 4’’ and 5’’ plates. …of S. precaria have two…

Response - Corrected

Comment 79: l 400 …of the 5’’ plate (i.e., contacting the 2a plate or not), despite…

Response - Corrected

Comment 80: l 402 …was not observed by…

Response - Corrected

Comment 81: l 409 …of S. precaria is the same…

Response - Corrected

Comment 82: l 411 please note that you used the name S. trochoidea till now, so you have to continue to use this name and not S. acuminata. Anyway, I recommend using across the whole ms the new combination Scrippsiella acuminata instead of the heterotypic synonym of S. trochoidea.

Response – We have clarified that.

Comment 83: l 416 … is consistent…

Response - Corrected

Comment 84: l 426 …of harmful microalgae…

Response - Corrected

Comment 85: l 431 …with pointed spines…

Response - Corrected

Comment 86: l 432 …spiny cysts sometimes showing capitate ends… References

Response - Corrected

Comment 87: l 449 … Scrippsiella acuminata…

Response - Corrected

Round 2

Reviewer 2 Report

The comments were addressed properly. Minor language revision would be beneficial.